# Effects of Desiccation on Metamorphic Climax in *Bombina variegata*: Changes in Levels and Patterns of Oxidative Stress Parameters

**DOI:** 10.3390/ani11040953

**Published:** 2021-03-29

**Authors:** Tamara G. Petrović, Ana Kijanović, Nataša Kolarov Tomašević, Jelena P. Gavrić, Svetlana G. Despotović, Branka R. Gavrilović, Tijana B. Radovanović, Tanja Vukov, Caterina Faggio, Marko D. Prokić

**Affiliations:** 1Department of Physiology, Institute for Biological Research “Siniša Stanković”, National Institute of Republic of Serbia, University of Belgrade, Bulevar despota Stefana 142, 11060 Belgrade, Serbia; tamara.petrovic@ibiss.bg.ac.rs (T.G.P.); jelena.gavric@ibiss.bg.ac.rs (J.P.G.); despot@ibiss.bg.ac.rs (S.G.D.); perendija@ibiss.bg.ac.rs (B.R.G.); tijana@ibiss.bg.ac.rs (T.B.R.); marko.prokic@ibiss.bg.ac.rs (M.D.P.); 2Department of Evolutionary Biology, Institute for Biological Research “Siniša Stanković”, National Institute of Republic of Serbia, University of Belgrade, Bulevar despota Stefana 142, 11060 Belgrade, Serbia; ana.kijanovic@ibiss.bg.ac.rs (A.K.); natasha@ibiss.bg.ac.rs (N.K.T.); tvukov@ibiss.bg.ac.rs (T.V.); 3Department of Chemical, Biological, Pharmaceutical and Environmental Sciences, University of Messina, 98166 Messina, Italy

**Keywords:** amphibian larvae, antioxidant system, oxidative damage, pond drying, metamorphosis, yellow-bellied toad

## Abstract

**Simple Summary:**

Global warming alters patterns of precipitation and drought, which are important factors in the survival of amphibian populations. Metamorphosis is affected by environmental changes; this is especially true of metamorphic climax, the crucial stage of amphibian development that is accompanied by significant morphological, physiological and behavioral adaptations necessary for the transition to a terrestrial habitat. This study investigated naturally occurring changes in the cellular oxidative status (antioxidant system and oxidative damage) of yellow-bellied toad larvae during this phase, and how exposure to exogenous factors such as desiccation affected them. Our results revealed clear changes in the antioxidant system’s (AOS) response and the levels of oxidative damage during metamorphic climax, with the highest response and damage observed at the end stage. Decreasing water levels during larval development altered the components of the AOS and increased oxidative damage, resulting in increased oxidative stress. The knowledge gained from this study could contribute to a better understanding of the oxidative stress that larvae experience during this critical stage of development, and the consequences of global warming—such as water loss—on amphibians.

**Abstract:**

In this paper, we examined how the oxidative status (antioxidant system and oxidative damage) of *Bombina variegata* larvae changed during the metamorphic climax (Gosner stages: 42—beginning, 44—middle and 46—end) and compared the patterns and levels of oxidative stress parameters between individuals developing under constant water availability (control) and those developing under decreasing water availability (desiccation group). Our results revealed that larvae developing under decreasing water availability exhibited increased oxidative damage in the middle and end stages. This was followed by lower levels of glutathione in stages 44 and 46, as well as lower values of catalase, glutathione peroxidase, glutathione S-transferase and sulfhydryl groups in stage 46 (all in relation to control animals). Comparison between stages 42, 44 and 46 within treatments showed that individuals in the last stage demonstrated the highest intensities of lipid oxidative damage in both the control and desiccation groups. As for the parameters of the antioxidant system, control individuals displayed greater variety in response to changes induced by metamorphic climax than individuals exposed to desiccation treatment. The overall decrease in water availability during development led to increased oxidative stress and modifications in the pattern of AOS response to changes induced by metamorphic climax in larvae of *B. variegata*.

## 1. Introduction

Global climate change and subsequent processes are considered responsible for decreasing populations of various wild species [1,2]. Amphibian species are recognized as a particularly globally threatened vertebrate taxa; we are witnessing worldwide threats faced by batrachofauna from habitat modification, environmental pollution, global warming, emerging pathogens and invasive species, all of which present a dramatic challenge for both science and society [3]. Due to their ectothermic nature, semipermeable skin and complex life cycles, amphibians are strongly dependent on specific temperature ranges, moisture, and precipitation—factors which are fundamentally affected by the process of global warming [3,4]. There is evidence that climate change, especially temperature-related change, can alter reproductive phenology, body condition and sensitivity to disease in amphibians [5]. The shortening of wetland hydroperiods has resulted in an increased frequency of reproductive failure for many amphibian species that use temporary or even ephemeral aquatic habitats for reproduction [5,6]. In particular, pond drying is affecting the development of anuran larvae [7]. It can lead to changes in water temperature, larval densities, food availability and host-pathogen or predator-prey interactions—all of which are directly linked to survival [8]. Larvae exposed to desiccation pressure need to make an ontogenetic niche shift before a pond dries up [6]. This shift, however, carries trade-offs that may have short- and long-term costs [9]. Costs could be seen in various morphological and physiological traits, e.g., reduced size at metamorphosis, leading to smaller maturation size, lower fecundity, reduction of hindlimb length, accelerated development, higher metabolic rates and alterations in oxidative metabolism and the immune system [6,7,8,10,11,12,13].

Anuran metamorphosis is a series of transformations that enables the transition from aquatic to terrestrial life. It is characterized by rapid changes in body shape and function [14,15]. The continuous process of metamorphosis is divided into three main phases: premetamorphosis, prometamorphosis and metamorphic climax [16]. In premetamorphosis (covering Gosner stages (GS) 25–35) larvae begin free-swimming and are in direct contact with their surroundings. During the prometamorphic phase (GS 35–41), larvae undergo intense growth and feeding rates, accumulating the energy necessary for further development. However, in the last phase, the metamorphic climax (GS 42–46), larvae reduce their growth, body mass and food intake [17]. This is the phase with the most far-reaching changes. As a critical phase in determining individual survival and population dynamics, it is the focus of ecological and toxicological studies [18] that consider the coordinated maturation and remodeling of organs, appearance of limbs and reduction of gills and tail, all of which are followed by balanced processes of cell death and proliferation [19]. The activation of the hypothalamic-pituitary-thyroid (HPT) axis during these processes is followed by numerous behavioral, physiological and biochemical changes—and modifications of oxidative metabolism in particular [11,20]. Johnson, et al. reported enhanced production of reactive oxygen species (ROS) and reactive nitrogen species (RNS) in this phase [21]. It was suggested that reactive species play an important regulatory role in the process of metamorphic organ changes [21,22]. ROS activate the apoptosis-linked pathway that underlies mechanisms such as tail resorption and remodulation of the gastrointestinal system in larvae [23,24]. However, overproduction of ROS can lead to oxidative damage of biomolecules (nucleic acids, proteins and lipids). The antioxidant system (AOS) assumes the main role in controlling ROS and RNS production and managing developmental processes [22,25,26,27]. It is a multifaceted system, containing various exogenous and endogenous antioxidants that act as enzymatic or nonenzymatic components [28].

In the present study, we analyzed changes in oxidative stress parameters (antioxidant system and oxidative lipid damage) at the beginning, middle and end of metamorphic climax in *Bombina variegata*. Changes in oxidative status (ROS, lipid peroxide (LPO) concentrations and AOS) under metamorphic climax play an important role in organ remodeling and preparing tadpoles for terrestrial life. Naturally occurring changes in this status may be additionally affected by exogenous/environmental factors such as pond desiccation. The effects of water loss on oxidative stress in anurans have been investigated only in juvenile and prometamorphic individuals [11,12], while information about oxidative stress during metamorphic climax is missing. Therefore, we examined how desiccation affects body size and physiological traits at this crucial phase of anuran development. We investigated whether water loss might alter the sensitive balance between the formation of different ROS and the AOS, which would result in increased oxidative stress in larvae developing under decreasing water availability in comparison to larvae growing in an unchanging water level.

## 2. Materials and Methods

### 2.1. Experimental Design

On the same day in April 2019, we collected three clutches of *Bombina variegata* from different forest ephemeral rain ponds at Fruška Gora National Park. The eggs were transported to the laboratory of the Department of Evolutionary Biology at the Institute for Biological Research Siniša Stanković in Belgrade. There, they were kept in separate plastic boxes filled with 2 L of dechlorinated tap water until the individuals reached developmental stage 25 (GS 25, free swimming and feeding [16]). The temperature during the experiment was maintained at 20 °C with a natural photoperiod (14L/10D h). At this stage, GS 25 individuals from each clutch were randomly assigned to the following treatment groups: (i) control group, maintained at a constant water level (water volume: 10 L, water depth: 9.4 cm) and (ii) desiccation group, exposed to decreasing water levels. In the latter group, the water level at the start of the experiment was 10 L (water depth: 9.4 cm) and was decreased every 4th day by 20%, starting on day 5 and continuing until day 25, after which the water volume was kept constant at 1.3 L (water depth: 1.25 cm) (Figure 1) [29]. Each treatment group contained 3 experimental units (plastic containers: 42 × 32 × 21.5 cm) with 10 individuals per container. The water was changed every 4th day, together with the position of the containers (to minimize the possible effects of the position). Tadpoles were fed ad libitum every second day with the same amount of commercial fish food (two tablets of Tetra TabiMin^®^ (Tetra GmbH, Melle, Germany) per container). The development of individuals was checked every day. After emergence of forelimbs at GS 42, a wet terrestrial habitat was provided in the container for metamorphic individuals by tilting the box to allow both wet and dry areas. For the examination of oxidative stress parameters, we randomly took at least 3 individuals from each container at the beginning (GS 42), middle (GS 44) and end (GS 46) stages of metamorphic climax. For determination of body size, we took dorsal photographs of the live tadpoles with a Sony DSC-F828 digital camera (24-bit color and 3264 × 2448 pixel resolution; Sony Corp., Tokyo, Japan). The photos were then analyzed by tpsDig2 software [30]. Tadpole body size was calculated as the distance from the snout tip to the end of the tail for GS 42 and GS 44, and as the distance from the snout tip to the end of the outer edge of the cloaca for GS 46. We also recorded the number of days from the start of the experiment (GS 25) to every stage of interest in the metamorphic climax (larval developmental time). Tadpoles were then euthanized by placement in liquid nitrogen [31].

The collection of anuran eggs for the experiment was approved by the Ministry of Environmental Protection of the Republic of Serbia (permit no. 353-01-83/2019-04). The Animal Ethical Committee of the Institute for Biological Research Siniša Stanković, University of Belgrade, issued the approval for the experimental procedure (decision no. 04-5/19). We followed all European Directives (2010/63/EU) on the protection of animals used for experimental and other scientific purposes, as well as the ARRIVE guidelines and Code of Practice for the Housing and Care of Animals used in Scientific Procedures.

### 2.2. Sample Processing

Whole bodies of larvae were finely chopped and mixed to obtain as much homogenous material as possible. After that, 0.2 g was taken to determine the concentration of thiobarbituric acid reactive substance (TBARS) as a standard marker for LPO (lipid peroxidation); the rest was used for determination of antioxidant parameters. Tadpoles were individually homogenized with an Ultra Turrax homogenizer T-18 (IKA-Werk, Staufen, Germany) at a 1:5 ratio in an ice cold 25 mM sucrose buffer (pH 7.4) containing 10 mM Tris-HCl and 5 mM EDTA. The homogenates were sonicated with an ultrasonic homogenizer (Sonopuls HD 2070, Bandelin electronic, Berlin, Germany) for 30 s at 10 kHz. A part of the sonicate was taken for measurement of total reduced glutathione (GSH) concentration, while the rest was centrifuged in an L7-55 ultracentrifuge (Beckman, Brea, CA, USA) at 100,000× *g* at 4 °C for 90 min [32]. The supernatants were used for measuring AOS parameters.

### 2.3. Biochemical Parameters

The Lowry method was used to determine protein concentration, with bovine serum albumin (BSA) as standard [33]. The activity of superoxide dismutase (SOD) was measured as the autoxidation of adrenaline to adrenochrome, according to the procedure of Misra and Fridovich [34]; absorbance was monitored at 480 nm. The activity of catalase (CAT) was quantified by the Claiborne [35] method, which involves the measurement of hydrogen peroxide breakdown at 240 nm. Glutathione peroxidase (GSH-Px) activity was assayed in accordance with the protocol given by Tamura et al. [36]. The method is based on the oxidation of nicotinamide adenine dinucleotide phosphate (NADPH) with t-butyl hydroperoxide as substrate. Glutathione reductase (GR) activity determination was based on the ability of GR to reduce glutathione disulphide (GSSG) to GSH using NADPH as a substrate [37]. The reaction of the sulfhydryl (SH) group of GSH with 1-chloro-2,4-dinitrobenzene (CDNB) was used to determine glutathione-S-transferase (GST) activity [38]. The activities of all glutathione-related enzymes (GSH-Px, GR and GST) were recorded at 340 nm. All enzyme activities were expressed in U/mg protein.

The Griffith [39] method, based on 5,5′-dithio-bis-(2-nitrobenzoic acid) (DTNB) enzymatic recycling, was used to measure the concentration of total GSH; values of GSH were expressed in nmol/g tissue. Sulfhydryl—SH group concentration was measured at 412 nm after incubation of tissue extracts with DTNB, and expressed in µmol/g tissue [40]. Lipid peroxidation, or thiobarbituric acid-reactive substance (TBARS) estimation, was measured by the method described by Rehncrona et al. [41] at 532 nm and expressed in nmol TBARS/mg tissue.

A detailed description of the utilized biochemical methods is presented in the Appendix A. All biochemical parameters were measured at 25 °C using a UV-VIS spectrophotometer (UV-1800, Shimadzu, Japan). All chemicals were obtained from Sigma (St. Louis, MO, USA).

### 2.4. Statistical Analyses

We tested outliers (Grubb’s test), the homogeneity of variances (Levine’s test) and assumptions of normality according to the Kolmogorov–Smirnov test. As the preliminary analyses did not reveal any significant differences for each oxidative stress parameter between experimental units (containers) for each stage during the treatments (Appendix A), the factor container was not retained in the analyses. Therefore, pooling individuals from containers increased the number of samples per factor (treatment and stage).

One-way ANOVA was used to examine differences in body size and larval developmental time (DT) between treatments for each stage. Further, factorial ANOVA was applied to examine possible differences between two independent variables—treatment (decreasing water level for the desiccation group, and constant water level for the control) and stages (42, 44 and 46)—and their interaction with oxidative stress parameters. For parameters for which a significant interaction between factors (treatment x stage) was observed, we performed pairwise multiple comparisons with Tukey’s test. For parameters that displayed significant differences for each factor, the post hoc Tukey HSD (honestly significant difference) test was used. Statistical analyses were performed using STATISTICA 8.0 [42]—except pairwise multiple comparisons, which were calculated in XLSTAT, Ver. 2014.5.03 [43].

## 3. Results

The values for body length (BL) showed that individuals from the desiccation group had significantly lower body lengths at each examined stage compared to individuals that developed under constant water availability (stage 42: BL from desiccation individuals 37.04 ± 0.60 mm, BL from control individuals 41.94 ± 0.71 mm, df = 1, F = 35.11, *p* < 0.00001; stage 44: BL from desiccation individuals 28.17 ± 1.24 mm, BL from control individuals 33.35 ± 1.12 mm, df = 1, F = 9.61, *p* = 0.00382; stage 46: BL from desiccation individuals 14.35 ± 0.30 mm, BL of control individuals 16.81 ± 0.26 mm, df = 1, F = 39.02, *p* < 0.00001). However, the treatment did not have a significant effect on the DT at each stage (the DT of stage 42 desiccation individuals was 41.18 ± 2.82 days, compared to the DT of control individuals: 39.88 ± 0.92 days, df = 1, F = 1.72, *p* = 0.20599; the DT of stage 44 desiccation individuals was 42.77 ± 3.00 days, compared to the DT of control individuals: 42.55 ± 2.63 days, df = 1, F = 0.05, *p* = 0.81492; the DT of stage 46 desiccation individuals was 54.73 ± 3.39 days, compared to the DT of control individuals: 52.80 ± 1.65 days, df = 1, F = 3.93, *p* = 0.05709).

The results for factorial ANOVA showed significant differences for CAT, GSH-Px, GSH and LPO for the factors treatment and stage. Significant differences between stages were only recorded for SOD and GR activities and SH group concentration. Significant interactions (treatment × stages) were observed for CAT, GSH-Px, GST and SH groups (Table 1). All combinations of all post hoc analyses of the significant interactions between treatment and stages for tested parameters are given in Appendix A.

Differences between desiccation treatment and matching control at each stage are presented in Figure 2 and Figure 3. The results for larvae at stage 42 did not reveal any significant differences between the examined treatments. At stage 44, individuals exposed to desiccation had a lower concentration of GSH and a higher concentration of LPO with respect to the matching control. Larvae exposed to decreasing water at stage 46 had lower GSH and higher LPO values, as well as lower values for CAT, GSH-Px, GST and SH groups (Figure 2 and Figure 3) compared to matching controls. SOD activity did not differ significantly between treatments.

Differences in developmental stages and accompanying changes in oxidative stress parameters in larvae subjected to decreasing water availability are presented in Figure 2 and Figure 3. The highest SOD activity was recorded in larvae at stage 42 in relation to stages 44 and 46. Individuals at stage 44 had a higher concentration of SH groups than individuals at stage 42. Comparison of stage 46 with stages 44 and 42 showed higher activity for GR, a higher concentration of LPO and lower CAT activity. Individuals at stage 46 also had a higher concentration of GSH in comparison to individuals at stage 42. No significant differences between stages were observed for GSH-Px and GST activities.

Differences in developmental stages and accompanying parameters in control larvae are presented in Figure 2 and Figure 3. Larvae at developmental stage 42 displayed higher CAT activity than individuals at stage 44. The greatest number of significant differences was reported for oxidative stress parameters in tadpoles at stage 46. Results revealed that they had higher values of GR, GSH, LPO, GSH-Px and SH groups than the other stages. They also displayed higher CAT and GST activities when compared to individuals from GS 44. The only parameter that did not display significant differences between stages was SOD activity.

## 4. Discussion

Exposure of *Bombina variegata* tadpoles to decreasing water availability (desiccation conditions) resulted in smaller body size but did not affect the pace of development. Some anuran species are incapable of increasing their developmental rates in response to drying conditions, as they already develop near their maximum physiological capacities [44]. Smaller body sizes without accelerated developmental rates at the end of the metamorphosis of *B. variegata* individuals during a period of desiccation when compared to individuals developing under steady water availability were reported by Sinsch et al. [7] and Böll [45]. The body size can reflect on various processes and fitness traits in amphibian larvae, including metabolism, resistance to pathogens, feeding behavior, locomotion, escape from predators [6,20]. In adults, body size can also affect mortality rate, age and size at first reproduction or fecundity [6,13].

Body size is considered to be one of the crucial traits for metamorphic climax. During this phase, oxygen consumption decreases, and individuals enter obligatory fasting, lose body mass and rely only on internal energy sources [46,47]. Any more pronounced alterations in energy balance throughout this demanding phase reflect on other biological processes [48,49,50]—and consequently on the oxidative status of larvae. In our study, tadpoles reared under declining water conditions displayed higher levels of oxidative damage in the middle and end stages of metamorphosis in comparison to individuals reared under constant water conditions. Increased oxidative damage of lipids in individuals exposed to desiccation could be interpreted as the effect of crowding stress mediated by the hypothalamic-pituitary-interrenal (HPI) axis [11,51] and through resource- or energy-based models [52,53]. The activation of the HPI axis in individuals exposed to desiccation results in elevated corticosterone levels—above the levels expected during this life stage—which induce the cellular reduction-oxidation (redox) system, increasing ROS production and oxidative stress [12,51]. According to the energy-based model, individuals with a smaller body size at the onset of metamorphosis would be at a significant physiological disadvantage in relation to larger tadpoles because they have completed metamorphosis less efficiently and with proportionally greater energy expenditure [48,54,55]. Ruthsatz et al. [48] showed that environmental stressors alter energy allocation in larvae of *Rana temporaria*, leaving less energy for development during metamorphic climax in comparison to control animals. Trade-off energy allocation between normal development and immune system development was also observed in larvae in a drying pond [8]. Desiccation-stressed larvae of *B. variegata* had a lower AOS response, and therefore a disabled system, incapable of sustaining increased ROS production. This resulted in greater oxidative stress, as observed in larvae facing pond drying. We assumed that the lower AOS response was due to the allocation of energy from investment in a costly antioxidant defense to maintain processes necessary for survival. In support of this hypothesis, we observed lower GSH concentrations in the desiccation group at stages 44 and 46 when compared to the control. The obligate fasting (due to the gastrointestinal and feeding apparatus remodeling) observed during metamorphic climax can affect the synthesis of GSH and proteins containing cysteine in SH groups, as they directly depend on food intake or the metabolism of dietary methionine [56,57]. At the end of metamorphosis (stage 46), individuals from the desiccation group needed to allocate more additional energy than control individuals. As a result, the values of other AOS parameters continued to decrease, including the activities of H_2_O_2_ scavenging enzymes (CAT and GSH-Px), the activity of GST (which is involved in lowering the peroxidation process) and the concentration of SH groups (compared to control individuals). This points to higher concentrations of ROS (H_2_O_2_) and increased production of LPOs. A previous study showed that limited energy availability in short-term fasting amphibian larvae also produces an inadequate AOS response—in particular, CAT, GSH-Px and GSH activities—to cope with higher ROS production and lipid oxidative damage [50]. The inability of larvae to invest in the AOS and immune systems in the absence or allocation of energy under stressful conditions (fasting, migration, accelerated development, reproduction) were also reported for various organisms [57,58,59]. *Pelobates cultripes* tadpoles that developed under declining water conditions displayed accelerated development and reduced growth [12], which in juvenile individuals induced higher activity of GSH-Px and GR triggered by increased ROS production [12].

This study revealed that concentrations of LPO were highest in both treatments at stage 46, with the higher concentration in the desiccation group, as a result of the accumulation of ROS from previous stages. Oxidative stress may be an inescapable consequence in the metamorphic climax in tadpoles due to the role of ROS in controlling metamorphosis—and because of enhanced physiological and metabolic processes [11,15,21,22,24,60]. However, the pattern of the AOS response differed between states of stable and decreasing water availability during all three stages of metamorphic climax. The results for AOS parameters in tadpoles experiencing simulated pond drying conditions revealed a low number of significant differences between stages, pointing to insufficient adaptive changes in the AOS during the different stages. In contrast, larvae developing under constant water availability exhibited more pronounced differences for a plurality of AOS parameters, with the highest values observed at the end stage of metamorphosis. We have reported higher values for the parameters of the GSH system (GSH, GR and GSH-Px), together with higher concentrations of SH groups, when compared to the beginning and middle stages. The higher concentration of GSH (which is capable of removing different forms of ROS) and GR (which reduces the oxidized form of GSH) [28] would protect the cells of individuals at the end of metamorphosis from further increased oxidative damage. GSH also acts as a redox buffer involved in determining cell fate (proliferation and apoptosis) [61] in the process of anuran tail regression [22]. GSH is a cofactor for GSH-Px activity that, with CAT, is involved in the removal of H_2_O_2_; in effect, it is a signaling molecule [62]. The better response of the AOS observed in control individuals at stage 46 was the the result of lowered ROS, which are not required after organ remodeling reaches completion [21,22,24,60]. The stronger AOS response was also an adaptation to higher oxygen consumption in the terrestrial environment. Individuals at stage 46 completed metamorphic changes and tended to spend more time in terrestrial environments. Increased AOS parameters in response to the new environment were reported in juvenile individuals of *Pelophylax esculentus* in a natural population [15].

Desiccation is an ongoing global environmental change experienced by anuran larvae throughout development. It leads to smaller body size, lower AOS response, increased oxidative damage of lipids and oxidative stress. In addition, decreasing water availability affects the pattern of changes of the AOS at the early, mid and late stages of the metamorphic climax. Oxidative stress during early life stages directly affects normal cell functioning and causes progressive deterioration of tissues. However, the long-term consequences on fitness-related traits at individual and population levels require further study. Understanding how different environmental conditions can affect the physiology of anuran species, especially during sensitive developmental phases, can be crucial for their conservation. This study presented new information about oxidative stress as an inescapable consequence of metamorphic climax—and also of its impact on the GSH system (GSH, GR, GSH-Px and GST) and how it affects the metamorphic process in developing larvae.

## Figures and Tables

**Figure 1 animals-11-00953-f001:**
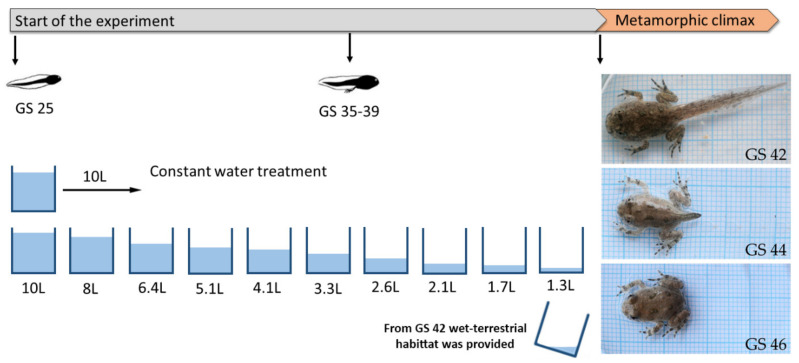
Experimental design.

**Figure 2 animals-11-00953-f002:**
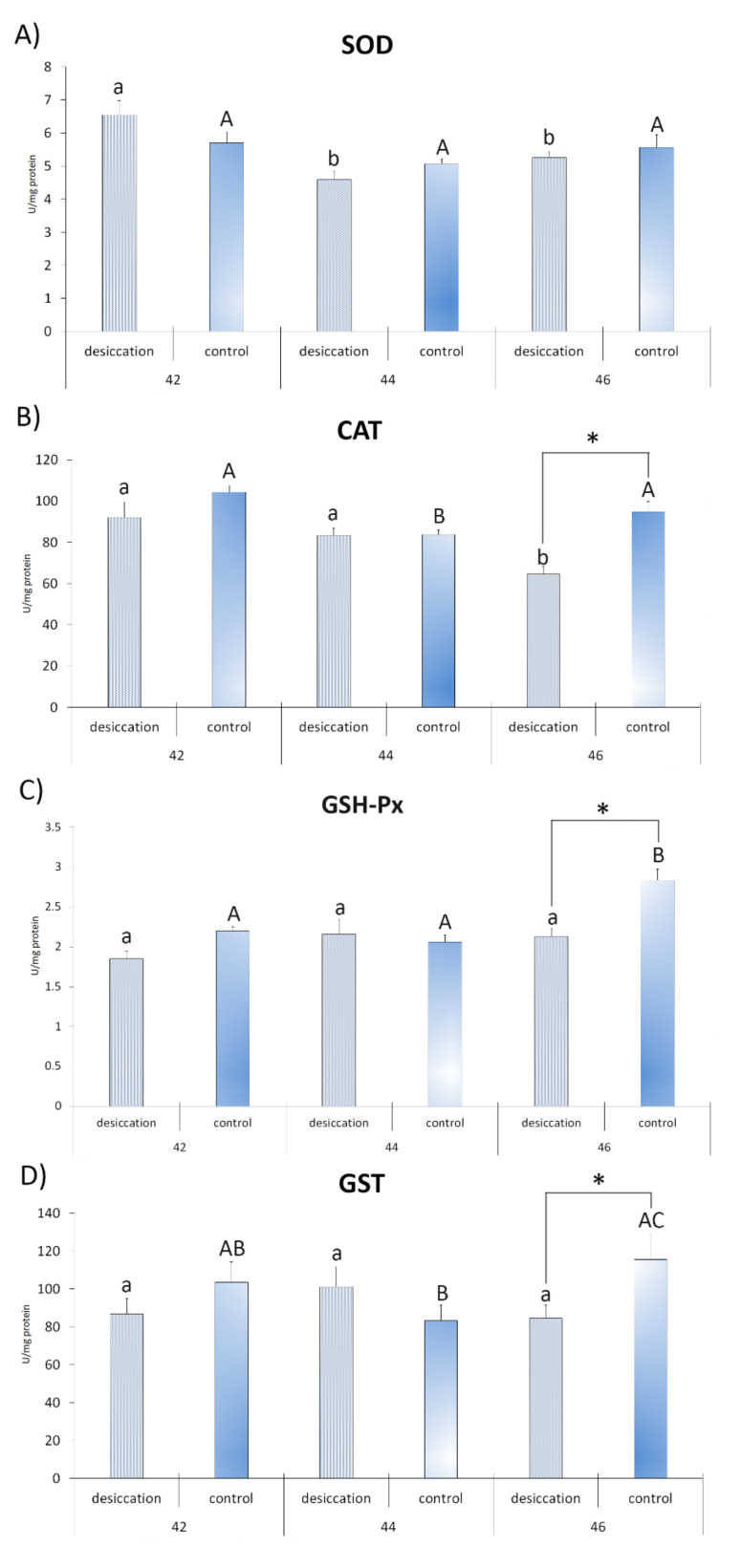
Antioxidant parameters (**A**)—superoxide dismutase (SOD); (**B**)—catalase (CAT); (**C**)—glutathione peroxidase (GSH-Px); and (**D**)—glutathione-S-transferase (GST)) in individuals from groups exposed to decreasing water availability (desiccation) and the control group, maintained at an unchanging water level (control) during metamorphic climax (Gosner stages 42, 44 and 46). * indicates significant differences between treatments (desiccation vs. control); different letters indicate significant differences between stages under each treatment (lower letters for desiccation, capital letters for the control).

**Figure 3 animals-11-00953-f003:**
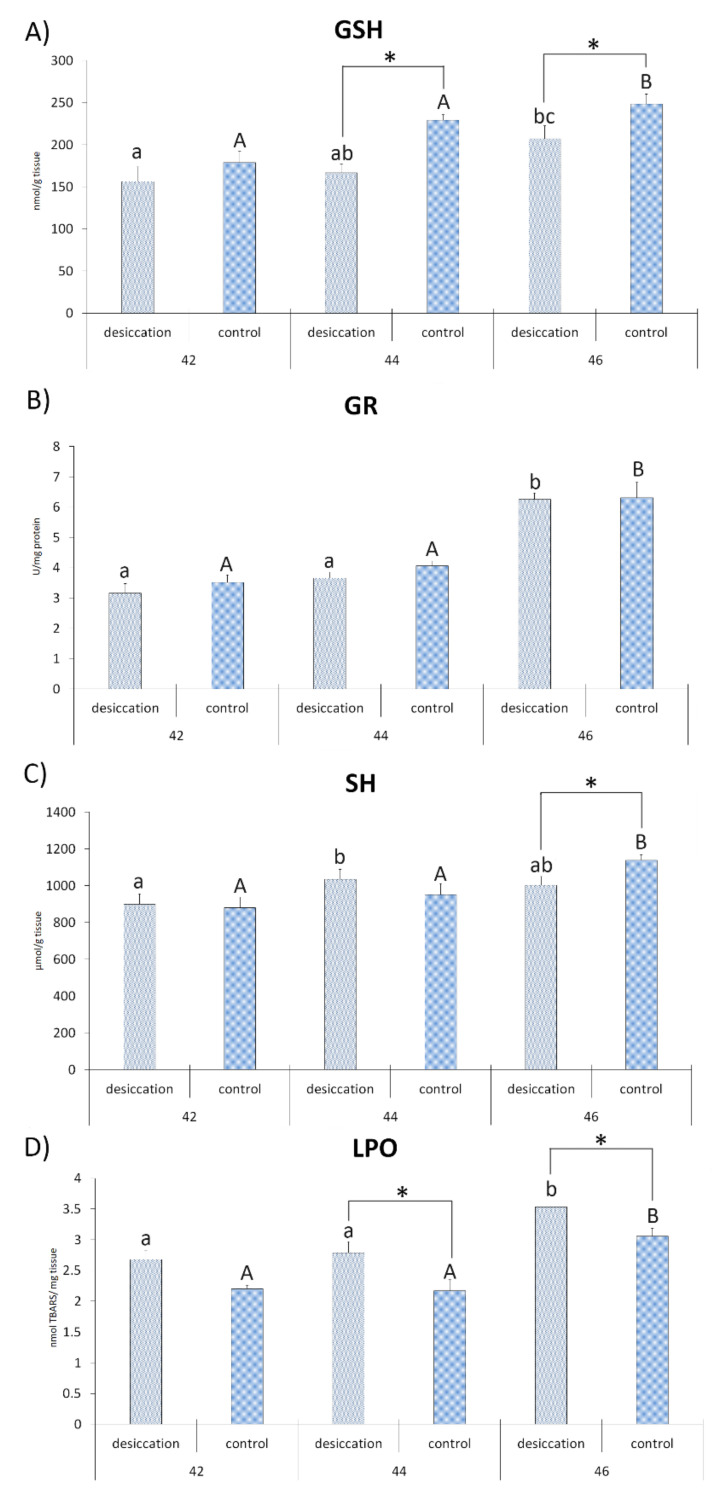
Oxidative stress parameters (**A**)—glutathione (GSH); (**B**)—glutathione reductase (GR); (**C**)—sulfhydryl (SH) groups; and (**D**)—lipid peroxide (LPO)) in individuals from groups exposed to decreasing water availability (desiccation) and a control group maintained at an unchanging water level during metamorphic climax (Gosner stages 42, 44 and 46).* indicates significant differences between treatments (desiccation vs. control); different letters indicate significant differences between stages under each treatment (lower letters for desiccation, capital letters for the control).

**Table 1 animals-11-00953-t001:** Results of factorial ANOVA for the comparison between the treatment (decreasing and constant water level), stage (Gosner stages 42, 44 and 46) and the interaction between the treatment and stage for oxidative stress parameters.

Parameter	Effects	df	F	*p*
SOD	treatment	1	0.004	0.94833
-	**stage**	**2**	**9.007**	**0.00036**
-	treatment × stage	2	2.573	0.08444
CAT	**treatment**	**1**	**15.944**	**0.00017**
-	**stage**	**2**	**8.975**	**0.00037**
-	**treatment × stage**	**2**	**6.210**	**0.00348**
GSH-Px	**treatment**	**1**	**8.953**	**0.00406**
-	**stage**	**2**	**8.113**	**0.00078**
-	**treatment × stage**	**2**	**7.127**	**0.00170**
GSH	**treatment**	**1**	**18.382**	**0.00006**
-	**stage**	**2**	**11.369**	**0.00006**
-	treatment × stage	2	1.410	0.25197
GST	treatment	1	1.421	0.23781
-	stage	2	0.323	0.72505
-	**treatment × stage**	**2**	**3.410**	**0.03944**
GR	treatment	1	0.919	0.34136
-	**stage**	**2**	**41.668**	**<0.00001**
-	treatment × stage	2	0.171	0.84298
SH	treatment	1	0.091	0.76290
-	**stage**	**2**	**7.638**	**0.00109**
-	**treatment × stage**	**2**	**3.392**	**0.04009**
LPO	**treatment**	**1**	**14.118**	**0.00054**
-	**stage**	**2**	**20.676**	**<0.00001**
-	treatment × stage	2	1.113	0.33852

Data in bold indicate significant differences, df: degrees of freedom.

## Data Availability

The data presented in this study are available on request from the corresponding author. The data are not publicly available as they are property of the Ministry of Education, Science and Technological Development of the Republic of Serbia.

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
