# Peer review of "Effects of Desiccation on Metamorphic Climax in *Bombina variegata*: Changes in Levels and Patterns of Oxidative Stress Parameters"

_animals, 2021, doi:10.3390/ani11040953_

Round 1

Reviewer 1 Report

Review on the document entitled:

Effects of desiccation on metamorphic climax in Bombina variegata: Changes in levels and patterns of oxidative stress parameters

Overall, I do find the work very interesting since not many have focused on the development of amphibians as far as the transitional state of metamorphosis. This study may thus provide further insight in comprehend how this transition is made and on how may constraint metamorphs reach adulthood.

Just some minor suggestions. Please see below:

Please complete figure number 1, with the day at which you started to provide the terrestrial habitat.

Please include some more information about the “terrestrial” habitat that you provide, provenience of the soil, etc.

Why did you not weight the organisms? Aside length, body mass could be an important endpoint.

It is possible to, at the time you euthanized the individuals, to check if they are females or males?  If yes, maybe this differentiation may provide some responses to the biomarker’s analysis.

In the graphics the * are not very clear…maybe if you place a small dash above the two columns you want to show differences (control and dessication) and then on top of the dash place the *

Author Response

Please find attached a revised version of the manuscript entitled: „Effects of desiccation on metamorphic climax in Bombina variegata: Changes in levels and patterns of oxidative stress parameters” by authors: Tamara G. Petrović , Ana Kijanović, Nataša Tomašević Kolarov, Jelena P. Gavrić, Svetlana G. Despotović, Branka R. Gavrilović, Tijana B. Radovanović, Tanja Vukov, Caterina Faggio and Marko D. Prokić.

We are grateful to reviewers for their comments, and journal for efficiency and professionalism. All suggestions are accepted and highlighted in the main text (with red color for reviewer 1 and green for reviewer 2)

************************

Our response to Reviewer #1:

Effects of desiccation on metamorphic climax in Bombina variegata: Changes in levels and patterns of oxidative stress parameters. Overall, I do find the work very interesting since not many have focused on the development of amphibians as far as the transitional state of metamorphosis. This study may thus provide further insight in comprehend how this transition is made and on how may constraint metamorphs reach adulthood.

Dear Reviewer 1, we appreciate your time and are very thankful for all comments on our work. Changes made according to your suggestions are highlighted with red color in the main text.

Just some minor suggestions. Please see below:

Please complete figure number 1, with the day at which you started to provide the terrestrial habitat.

Figure 1 was changed according to the Reviewer's suggestion (line 152).

Please include some more information about the "terrestrial" habitat that you provide, provenience of the soil, etc.

More information about terrestrial habitat was added in lines 133-134.

by tilting the box to allow both wet and dry arias”

Why did you not weight the organisms? Aside length, body mass could be an important endpoint.

We agree with the Reviewer that body mass data would be valuable information for this study but by using body length as a proxy for body mass (as they are highly correlated) we avoid further disturbance of individuals due to additional manipulations prior to stress parameters quantification.

It is possible to, at the time you euthanized the individuals, to check if they are females or males?  If yes, maybe this differentiation may provide some responses to the biomarker's analysis.

The majority of the yellow-bellied toads reach their sexual maturity after two or three winters (Anholt et al. 1997), therefore we were not able to determine the sex of examined individuals. In addition, it is not expected that sex can influence biomarkers in this life stage as it may in the adult stage.

Anholt, B., Barandun, J., & Reyer, H. U. (1997). Reproductive ecology of Bombina variegata: aspects of life history. Amphibia-Reptilia, 18(4), 347-355.

In the graphics the * are not very clear...maybe if you place a small dash above the two columns you want to show differences (control and desiccation) and then on top of the dash place the *

Figures were changed. We connected with line groups that differed significantly (control and desiccation) and added "*" on top of them (lines 261, 270). Hope that the figures are now clear.

Reviewer 2 Report

This is an interesting article where the authors look at the patterns of oxidative stress during anuran metamorphosis, and how these are affected by different desiccation rates. The topic is original and relevant, given that the Global Change is expected to increase unpredictability in precipitation regimes in many areas worldwide, so amphibians are likely to face fast desiccation with greater frequency.

The experiments seem sound, and the results fully support the conclusions. I have detected no major flaws in this manuscript. However, I did catch some errors, most of them concerning the style rather than the science itself. I don’t think the authors will find much difficulty in solving these issues. Below is a detailed list containing these mistakes.

Line 50: Maybe “Global warming” could be replaced by “Global change” here, a wider concept that includes, but is not limited to, the current increase in average temperature worldwide.

Line 50: The word “following” could be ambiguous here, as one could think that you attempt to start a list. Alternatives such as “cascading”, “subsequent” or “concomitant” could be better options.

Line 53: “facing” should be “faced by”.

Line 80: “a” before “phase” should be “the”.

Line 106: “are” should be replaced with “is”.

Line 108: Actually, you didn’t assume it. You checked or investigated whether that happened.

Lines 114-115: The word order in this sentence is awkward. Please, consider something like “we collected three clutches of Bombina variegata from different forest ephemeral rain pods”.

Lines 119-120: How was this natural photoperiod attained? Could natural light get in the room?

Line 129-131: What was that amount?

Line 135: Were these photos taken on live or dead tadpoles?

Lines 135-138: This phrasing is not smooth. Try something like “we took dorsal photographs of tadpoles with a Sony DSC-F828 digital camera (), which were analyzed for geometric morphometry with software tpsDig2”. Actually, the sentence is a bit confusing. Was geometric morphometry analyzed at all?

Line 192: How many outliers were detected?

Line 209: In the in-text result mentions and in the tables as well, degrees of freedom should be stated.

Lines 273-274: “It was shown that” could be deleted, which would result in a smoother sentence.

Line 298: “large” sounds more formal than “big”.

Line 299: A different wording would be clearer. Consider something like “proportionally greater energy expenditure”.

Line 302: Do you mean a trade-off involving energy allocation between these traits?

Line 309: “Obligation” should be “obligate”.

Line 310: “due to the”.

Line 351: “from” should be “at”, and “tend” should be “tended”.

Line 353: “in natural populations” or “in a natural population”, depending on the context.

Line 356: Delete “to”.

Line 357: “early, mid and late stages”.

Author Response

Dear Editor,                                                                                             March 24, 2021

Please find attached a revised version of the manuscript entitled: „Effects of desiccation on metamorphic climax in Bombina variegata: Changes in levels and patterns of oxidative stress parameters” by authors: Tamara G. Petrović , Ana Kijanović, Nataša Tomašević Kolarov, Jelena P. Gavrić, Svetlana G. Despotović, Branka R. Gavrilović, Tijana B. Radovanović, Tanja Vukov, Caterina Faggio and Marko D. Prokić.

We are grateful to reviewers for their comments, and journal for efficiency and professionalism. All suggestions are accepted and highlighted in the main text (with red color for reviewer 1 and green for reviewer 2)

Our response to Reviewer #2:

This is an interesting article where the authors look at the patterns of oxidative stress during anuran metamorphosis, and how these are affected by different desiccation rates. The topic is original and relevant, given that the Global Change is expected to increase unpredictability in precipitation regimes in many areas worldwide, so amphibians are likely to face fast desiccation with greater frequency.

The experiments seem sound, and the results fully support the conclusions. I have detected no major flaws in this manuscript. However, I did catch some errors, most of them concerning the style rather than the science itself. I don't think the authors will find much difficulty in solving these issues. Below is a detailed list containing these mistakes.

Dear Reviewer, we are very glad that you liked our paper and are thankful for detailed and quality suggestions, and the time dedicated to helping us to improve our manuscript. We do our best to respond to all your comments and change the manuscript according to them, all changes are highlighted in green color.

Line 50: Maybe "Global warming" could be replaced by "Global change" here, a wider concept that includes, but is not limited to, the current increase in average temperature worldwide.

"Global warming" was replaced with "Global change", we agree that Global change is a more appropriate term (line 50)

Line 50: The word "following" could be ambiguous here, as one could think that you attempt to start a list. Alternatives such as "cascading", "subsequent" or "concomitant" could be better options.

We choose "subsequent" as a better option to change "following" (line 50)

Line 53: "facing" should be "faced by".

Corrected (line 53)

Line 80: "a" before "phase" should be "the".

We have changed (line 80)

Line 106: "are" should be replaced with "is".

"are" was replaced with "is" (line 106)

Line 108: Actually, you didn't assume it. You checked or investigated whether that happened.

According to your suggestion, we replaced "Assumed" with "Investigated" (line 108)

Lines 114-115: The word order in this sentence is awkward. Please, consider something like "we collected three clutches of Bombina variegata from different forest ephemeral rain pods".

We rephrased the sentence as you wrote, thanks (lines 114-115)

Lines 119-120: How was this natural photoperiod attained? Could natural light get in the room?

Yes, natural photoperiod was obtained in a room with natural light (line 120).

Line 129-131: What was that amount?

Corrected (lines 130-131).

Line 135: Were these photos taken on live or dead tadpoles?

We used live tadpoles for photos (line 137) we added that fact to the main text.

Lines 135-138: This phrasing is not smooth. Try something like "we took dorsal photographs of tadpoles with a Sony DSC-F828 digital camera (), which were analyzed for geometric morphometry with software tpsDig2". Actually, the sentence is a bit confusing. Was geometric morphometry analyzed at all?

We have rewritten the mentioned sentence to obtain more clarity about the software that was used.  Geometric morphometry analyses were not performed at all, we just used tpsDig2 to quantify body length (lines 136-140).

Line 192: How many outliers were detected?

We did not found any outliers, just some data that were marked as further from the rest but not significantly (mostly for LPO)(lines 194-195)

Line 209: In the in-text result mentions and in the tables as well, degrees of freedom should be stated.

We added degrees of freedom – df for all comparison both in text and table (lines 216, 218, 220, 222, 224, 225, 237)

Lines 273-274: "It was shown that" could be deleted, which would result in a smoother sentence.

We agree, it was deleted (lines 280-281)

Line 298: "large" sounds more formal than "big".

Corrected (line 305)

Line 299: A different wording would be clearer. Consider something like "proportionally greater energy expenditure".

We accepted your suggestion and changed word ordering (lines 306)

Line 302: Do you mean a trade-off involving energy allocation between these traits?

Yes, and we stated so it can be clear (line 309)

Line 309: "Obligation" should be "obligate".

Done (line 317)

Line 310: "due to the".

Corrected (line 317)

Line 351: "from" should be "at", and "tend" should be "tended".

We replaced "from" with "at", and "tend" should be "tended" (line 359).

Line 353: "in natural populations" or "in a natural population", depending on the context.

Corrected to "in a natural population" (line 361)

Line 356: Delete "to".

Removed (line 365)

Line 357: "early, mid and late stages".

“beginning, middle and end stages” are replaced with "early, mid and late stages" it sounds better (lines 366-367)